# Gut microbiota-derived gamma-aminobutyric acid from metformin treatment reduces hepatic ischemia/reperfusion injury through inhibiting ferroptosis

Fangyan Wang[1†], Xiujie Liu[1,2,3†], Furong Huang[1†], Yan Zhou[4], Xinyu Wang[1], Zhengyang Song[1], Sisi Wang[1], Xiaoting Wang[1], Dibang Shi[5], Gaoyi Ruan[5], Xiawei Ji[5], Eryao Zhang[5], Zenglin Tan[1], Yuqing Ye[2,3], Chuang Wang[6], Jesse Zhu[2,3*], Wantie Wang[1*]

[1]Institute of Ischemia/Reperfusion Injury, School of Basic Medical Science, Wenzhou Medical University, Wenzhou, China; [2]Nottingham Ningbo China Beacons of Excellence Research and Innovation Institute, The University of Nottingham Ningbo, Ningbo, China; [3]Suzhou Inhal Pharma Co., Ltd., Suzhou, China; [4]Wenzhou Key Laboratory of Sanitary Microbiology, Wenzhou Medical University, Wenzhou, China; [5]Department of Gastroenterology, The Second Affiliated Hospital and Yuying Children's Hospital of Wenzhou Medical University, Wenzhou, China; [6]Medical School of Ningbo University, Ningbo University, Ningbo, China

*For correspondence:
jzhu@uwo.ca (JZ);
wwt@wmu.edu.cn (WW)

†These authors contributed equally to this work

**Abstract** Hepatic ischemia/reperfusion injury (HIRI) is a common and inevitable factor leading to poor prognosis in various liver diseases, making the outcomes of current treatments in clinic unsatisfactory. Metformin has been demonstrated to be beneficial to alleviate HIRI in recent studies, however, the underpinning mechanism remains unclear. In this study, we found metformin mitigates HIRI-induced ferroptosis through reshaped gut microbiota in mice, which was confirmed by the results of fecal microbiota transplantation treatment but showed the elimination of the beneficial effects when gut bacteria were depleted using antibiotics. Detailedly, through 16S rRNA and metagenomic sequencing, we identified that the metformin-reshaped microbiota was characterized by the increase of gamma-aminobutyric acid (GABA) producing bacteria. This increase was further confirmed by the elevation of GABA synthesis key enzymes, glutamic acid decarboxylase and putrescine aminotransferase, in gut microbes of metformin-treated mice and healthy volunteers. Furthermore, the benefit of GABA against HIRI-induced ferroptosis was demonstrated in GABA-treated mice. Collectively, our data indicate that metformin can mitigate HIRI-induced ferroptosis by reshaped gut microbiota, with GABA identified as a key metabolite.

## eLife assessment

This study presents a **valuable** finding on the impact of metformin-induced shifts in gut microbial community structure and metabolite levels for drug efficacy in a mouse model of liver injury. The current evidence supporting the claims of the authors is **solid**. This paper will be of broad interest to researchers across multiple disciplines, including the microbiome, liver disease, and pharmacology.

## Introduction

Hepatic ischemia/reperfusion injury (HIRI) remains a major cause of liver damage mainly following liver transplantation, complicated hepatectomy, hemorrhagic shock, and severe liver trauma (*Ye et al., 2020*; *Jiao et al., 2020*; *Liu et al., 2019*; *Liu et al., 2017b*). Traditional therapeutics have been developed and repurposed to treat HIRI, including ischemic preconditioning, machine perfusion technologies, gene interventions, and stem cell therapy (*Liu et al., 2021a*; *Czigany et al., 2020*; *Ke et al., 2006*; *Rowart et al., 2015*). Unfortunately, current treatments are not satisfactory due to the complicated pathogenesis of HIRI. Recently, a novel theory relating to the gut–liver axis has been well accepted, demonstrating that gut microbiota is vital for liver diseases, however, there is no strategy targeted this theory to alleviate HIRI in clinical practice.

The liver links with the intestine by portal circulation which allows the transfer of gut-derived products, not only nutrients, but also microbial metabolites and components (*Albillos et al., 2020*). As an important part of the gut–liver communication, the intestinal mucosal and vascular barriers limit the systemic dissemination of microbes and toxins while allowing nutrients to access the circulation and reaching the liver (*Tripathi et al., 2018*). Additionally, it is generally believed that liver injury can provoke intestinal mucosal damage and inflammation, further resulting in gut dysbiosis, which reversely induces a secondary attack on the liver (*Carbajo-Pescador et al., 2019*). Microbial agents, such as synbiotics and probiotics, have demonstrated to attenuate different liver diseases in animal studies (*Roychowdhury et al., 2019*; *Xu et al., 2021*), indicating that modulating gut microbes potentially prevents HIRI in clinical practice.

Recently, the hypoglycemic drug, metformin, has attracted intensive attention in its pleiotropic biological actions in various diseases, such as obesity, cancer, nuroinflammation, especially liver diseases (*Lv and Guo, 2020*; *Podhorecka et al., 2017*; *Kang et al., 2022*; *Xiong et al., 2019*). As evidences shown, the gut with high concentration of metformin become an important action site to reshape microbiota (*McCreight et al., 2016*). According to *Sun et al., 2018*, metformin was proven to decrease the abundance of the *B. fragilis* species and to increase the levels of bile acid and glycoursodeoxycholic acid in the gut, which inhibited intestinal farnesoid X receptor against obesity-related metabolic dysfunction in mice. Moreover, it is also demonstrated that metformin could attenuate sepsis-related liver injury by increasing the proportion of *Bifidobacterium*, *Muribaculaceae*, *Parabacteroides distasonis*, and *Alloprevitella* in rats (*Liang et al., 2022*). As for the HIRI, metformin has also shown the protective effect (*Jiang et al., 2021*), but the detailed role of gut microbiota remains unclear.

Although metformin showed protective effects on HIRI, the role and distinctions of reshaping microbes have yet to be determined. In the present study, we sought to elucidate that gamma-aminobutyric acid (GABA)-producing bacteria played a critical role for the protective effects of metformin against HIRI by using 16S rRNA and metagenomic sequencing to find the different features of gut microbiota. The finding was reconfirmed by the results of mice that were treated with metformin-reshaped fecal microbiota or oral antibiotics administration. Furthermore, GABA was found to show a protective effect on HIRI by intraperitoneally injection into mice in advance, while an increase of microbial GABA was observed in both mice and human experiments after metformin treatment. Evidenced by the improvement of HIRI, we hypothesize that metformin-reshaped gut microbiota elicits greater therapeutic responses for liver diseases and serve as a promising targeted therapeutic.

## Results

### Metformin significantly mitigates HIRI and reshapes gut microbiota in mice

In order to evaluate the protective effect of metformin on HIRI, survival experiment was employed in mice. The survival rate of HIRI mice increased from 41.67% to 90.00% after metformin intervention (*Figure 1a*), accompanied by hepatic morphological and histopathological improvement including the recovery of liver pathological structure, the decrease in red blood cell aggregation and consumption of liver glycogen (*Figure 1b*), as well as the decrease in serum activity of alanine transaminase (ALT) and aspertate aminotransferase (AST) (*Figure 1c*). Qualification of malondialdehyde (MDA) and glutathione (GSH) concentrations in the mice liver demonstrated that the presence of metformin significantly attenuated HIRI-induced oxidation (*Figure 1d*).

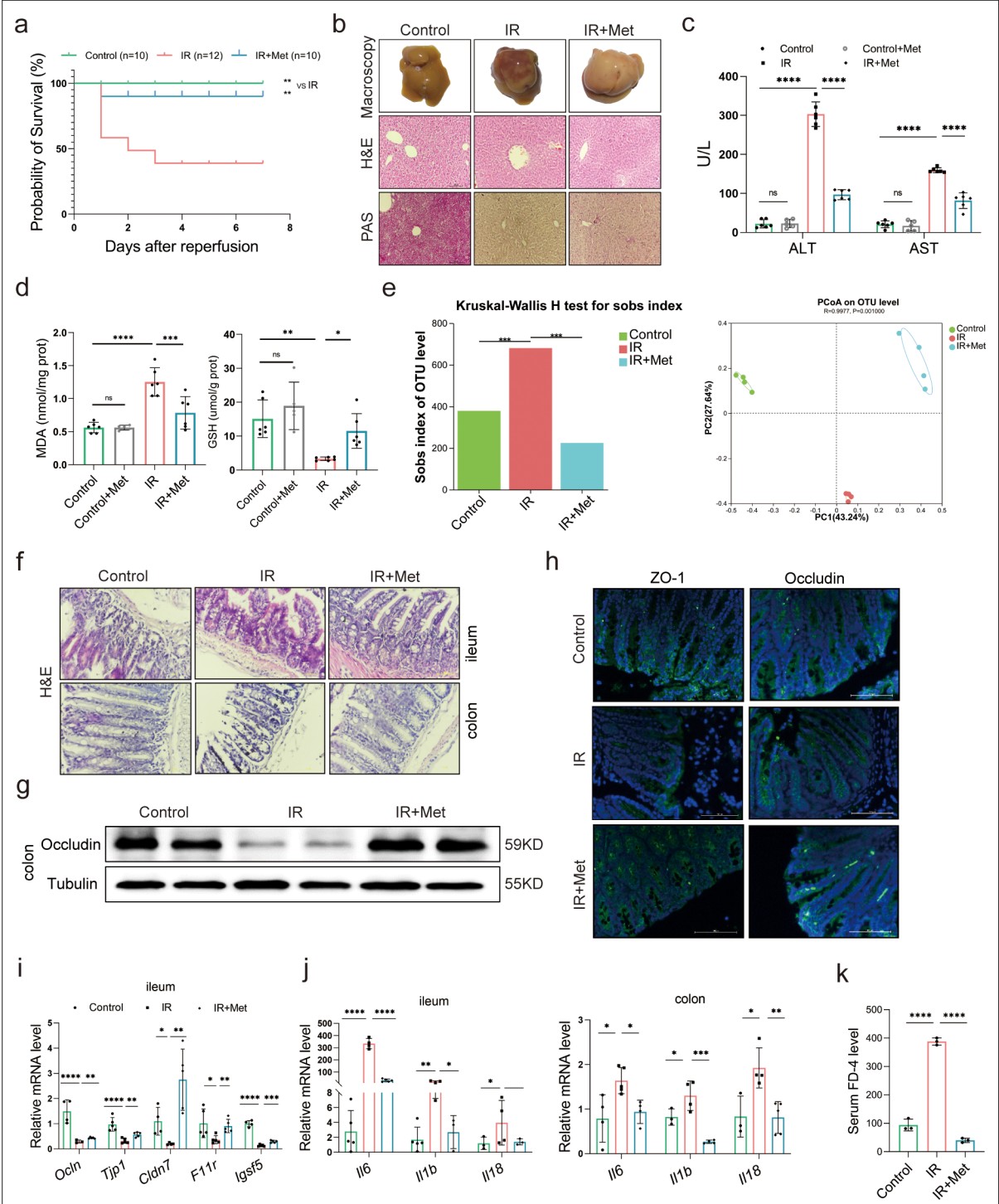

**Figure 1.** Metformin significantly mitigates hepatic ischemia/reperfusion injury (HIRI) by reshaping gut microbiota in mice. (**a**) Survival rate analysis ($n$ = 10–12/group), Kaplan–Meier was used to analyze statistical differences (Control vs IR, and IR + Met vs IR). (**b**) Morphological appearance, hematoxylin–eosin (H&E) and periodic acid-Schiff (PAS) staining of liver ($n$ = 3/group). (**c**) Serum alanine transaminase (ALT) and aspertate aminotransferase (AST) levels. (**d**) Levels of malondialdehyde (MDA) and glutathione (GSH) in liver tissue. (**e**) Sobs index, PCoA(principal co-ordinates analysis) ($n$ = 4/group). (**f**) H&E staining of ileum and colon ($n$ = 3/group). (**g**) Western blot of Occludin in colon ($n$ = 3/group). (**h**) IFC of ZO-1 and Occludin in colon ($n$ = 3/group). (**i**) Quantitative real-time PCR (qRT-PCR) analysis of *Ocln*, *Tjp1*, *Cldn7*, *F11r*, and *Igsf5* in ileum. (**j**) *Il6*, *Il1b*, and *Il18* in ileum and colon ($n$ = 3–5/group). (**k**) FD-4 level in mice serum ($n$ = 3). Data are expressed as mean ± standard deviation. One-way analysis of variance (ANOVA) was used to analyze statistical differences; *$p < 0.05$, **$p < 0.01$, and ***$p < 0.001$, $p$****<0.0001.

The online version of this article includes the following source data for figure 1:

*Figure 1 continued on next page*

*Figure 1 continued*

**Source data 1.** Original file for the Western blot analysis in *Figure 1g*.

**Source data 2.** PDF containing *Figure 1g* and original scans of the relevant Western blot analysis with highlighted bands and sample labels.

In order to clarify the critical role of gut microbiota in the pleiotropic actions of metformin (*Mueller et al., 2021*; *Lee et al., 2021*; *Liu et al., 2021b*), fecal samples were collected from the mice to perform 16S rRNA sequencing. It was found that the gut microbiota was substantially modulated in ischemia–reperfusion (IR) + metformin (Met) group, showing decreased abundance (alpha diversity) and reshaped distribution (*Figure 1e*). Next, as the microbiota–gut–liver axis theory indicates (*Milosevic et al., 2019*), HIRI-induced dysfunction of the gut barrier may aggravate liver damage by disrupting the gut microbiota. Hematoxylin–eosin (H&E) staining of the colon and ileum, mainly involved in gut barrier dysfunction, showed that the density and integrity of intestinal mucosa were strikingly reduced by HIRI, which was restored by metformin treatment (*Figure 1f*). The immunofluorescence analysis of colon tissues showed that the decreased tight junction protein ZO-1 and Occludin in HIRI-treated mice were reversed by metformin (*Figure 1g, h*). Similarly, Western blot analysis of Occludin in colon (*Figure 1g*) and the mRNA levels of *Ocln*, *Tjp1*, *Cldn7*, *F11r*, and *Igsf5* in ileum of HIRI mice were increased with metformin administration (*Figure 1i*). The inflammatory cytokine *Il6*, *Il1b*, and *Il18* were significantly upregulated in the ileum and colon received HIRI and were reduced by metformin treatment (*Figure 1j*). In addition, the serum FD-4 level in IR group was cut down by metformin (*Figure 1k*). These results suggested that metformin plays a critical role in restoring gut barrier function and alleviating HIRI.

## Metformin-reshaped fecal microbiota attenuates HIRI

In order to further confirm the role of metformin in reshaping the gut microbiota, microbes were transplanted into mice before inducing IR. As expected, the survival analysis of mice in the IR + fecal microbiota transplantation (FMT) group showed a similarly protective efficacy as the IR + Met group, as indicated by the significantly improved survival rate of IR mice (*Figure 2a*). This finding was validated by ALT and AST levels as well as H&E, periodic acid-Schiff (PAS), and dihydroethidium (DHE) staining (*Figure 2b, e*). FMT treatment could restore the GSH level but reduce the accumulation of MDA in hepatic tissues (*Figure 2c, d*). Furthermore, a decreased alpha diversity was observed in the FMT group, similar to IR + Met group (*Figure 2f*). Meanwhile, the attenuation of gut damage was showed by H&E staining, and the improvement in tight junction proteins ZO-1 and Occludin, and decreased HIRI-induced *Il1b*, *Il6*, and *Il18* (*Figure 2g–i*). Serum FD-4 level showed a sharply decrease in IR + FMT group (*Figure 2j*). To further identify the function of gut microbes, experiments were designed, and combination treatment of antibiotics (1 mg/ml penicillin sulfate, 1 mg/ml neomycin sulfate, 1 mg/ml metronidazole, and 0.16 mg/ml gentamicin) and metformin were employed for 1 week before IR treated. As results showed, the ALT, AST, and MDA detection in control group treated with Abx showed no obvious change to control group, indicating that the Abx had no influence on the liver. The survival rate of IR + Met + antibiotics (Abx) group was greatly reduced, while the levels of ALT, AST, MAD, and GSH were completely opposite to IR + Met group (*Figure 2k*). The results confirmed that gut microbiota is critical for the effect of metformin against HIRI.

## FMT alleviates HIRI-induced ferroptosis through reshaped fecal microbiota

Since the reperfusion-induced excessive oxidative stress takes place in iron-rich hepatocytes (*Bi et al., 2019*), experiments were designed to observe ferroptosis in HIRI. The ferroptosis inhibitor, DFO, significantly ameliorated the liver injury, suggesting that ferroptosis contributes to HIRI (*Supplementary file 3*). Hence, to find out the relation, ferroptosis-related indicators were also monitored in both IR + Met and IR + FMT groups. The increased accumulation of Fe after HIRI was significantly cut down by metformin and FMT treatment (*Figure 3a*). The quantitative real-time PCR (qRT-PCR) analysis showed that the HIRI-induced upregulation of ferroptotic parameters, *Acsl4*, *Slc7a11*, *Slc39a14*, and *Ptgs2*, were significantly decreased by metformin and its reshaped gut microbiota, respectively (*Figure 3b–e*). The results of Western blotting further confirmed the anti-ferroptosis of metformin and FMT by the remarkable reduction of ACSL4, TFR1, VDAC1, VDAC2, and VDAC3 levels and

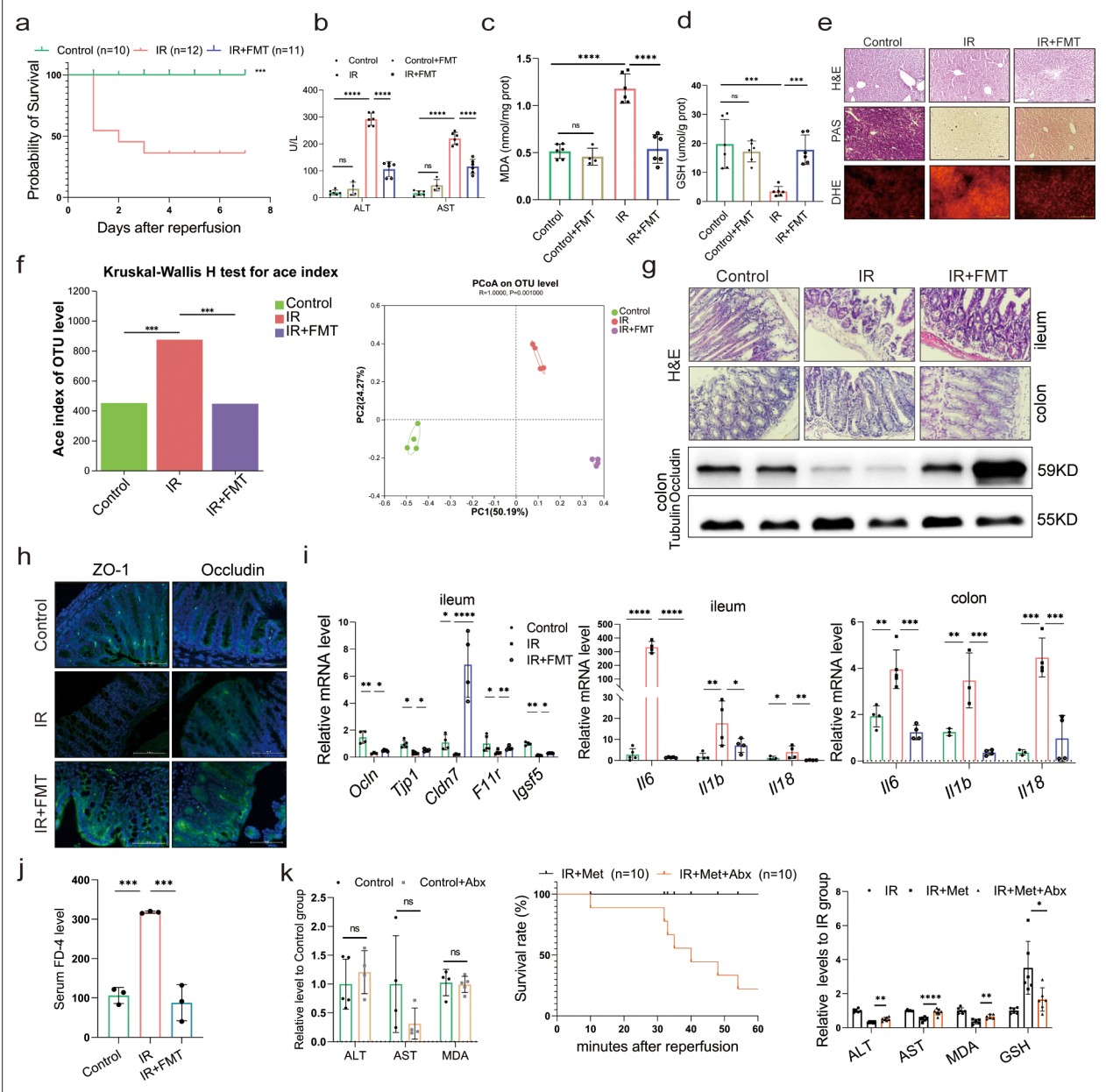

**Figure 2.** Metformin-reshaped fecal microbiota attenuates hepatic ischemia/reperfusion injury (HIRI). (**a**) Survival rate analysis (*n* = 10–12/group), Kaplan–Meier was used to analyze statistical differences. (**b**) Serum alanine transaminase (ALT) and aspertate aminotransferase (AST) levels. Levels of malondialdehyde (MDA) (**c**) and glutathione (GSH) (**d**) in liver tissue. (**e**) Hematoxylin–eosin (H&E), periodic acid-Schiff (PAS), and dihydroethidium (DHE) staining of liver (*n* = 3/group). (**f**) Alpha diversity and PCoA level of mice microbes. (**g**) H&E staining of ileum and colon and Western blot analysis of Occludin in colon (*n* = 3/group). (**h**) IFC of ZO-1 and Occludin in colon (*n* = 3/group). (**i**) Quantitative real-time PCR (qRT-PCR) analysis of *Ocln*, *Tjp1*, *Cldn7*, *F11r*, and *Igsf5* in ileum, and *Il6*, *Il1b*, and *Il18* in ileum and colon (*n* = 3–5/group). (**j**) FD-4 level in mice serum (*n* = 3). (**k**) ALT, AST, and MDA detection between Control and Control + Abx group; survival rate of ischemia–reperfusion (IR) + Met + Abx group (*n* = 10), Kaplan–Meier was used to analyze statistical differences; and the levels of serum ALT, AST, liver MDA, Fe, and GSH in IR + Met + Abx group. Data are expressed as mean ± standard deviation. One-way analysis of variance (ANOVA) was used to analyze statistical differences; *p < 0.05, **p < 0.01, and ***p < 0.001, p****<0.0001.

The online version of this article includes the following source data for figure 2:

**Source data 1.** Original file for the Western blot analysis in *Figure 2g*.

**Source data 2.** PDF containing *Figure 2g* and original scans of the relevant Western blot analysis with highlighted bands and sample labels.

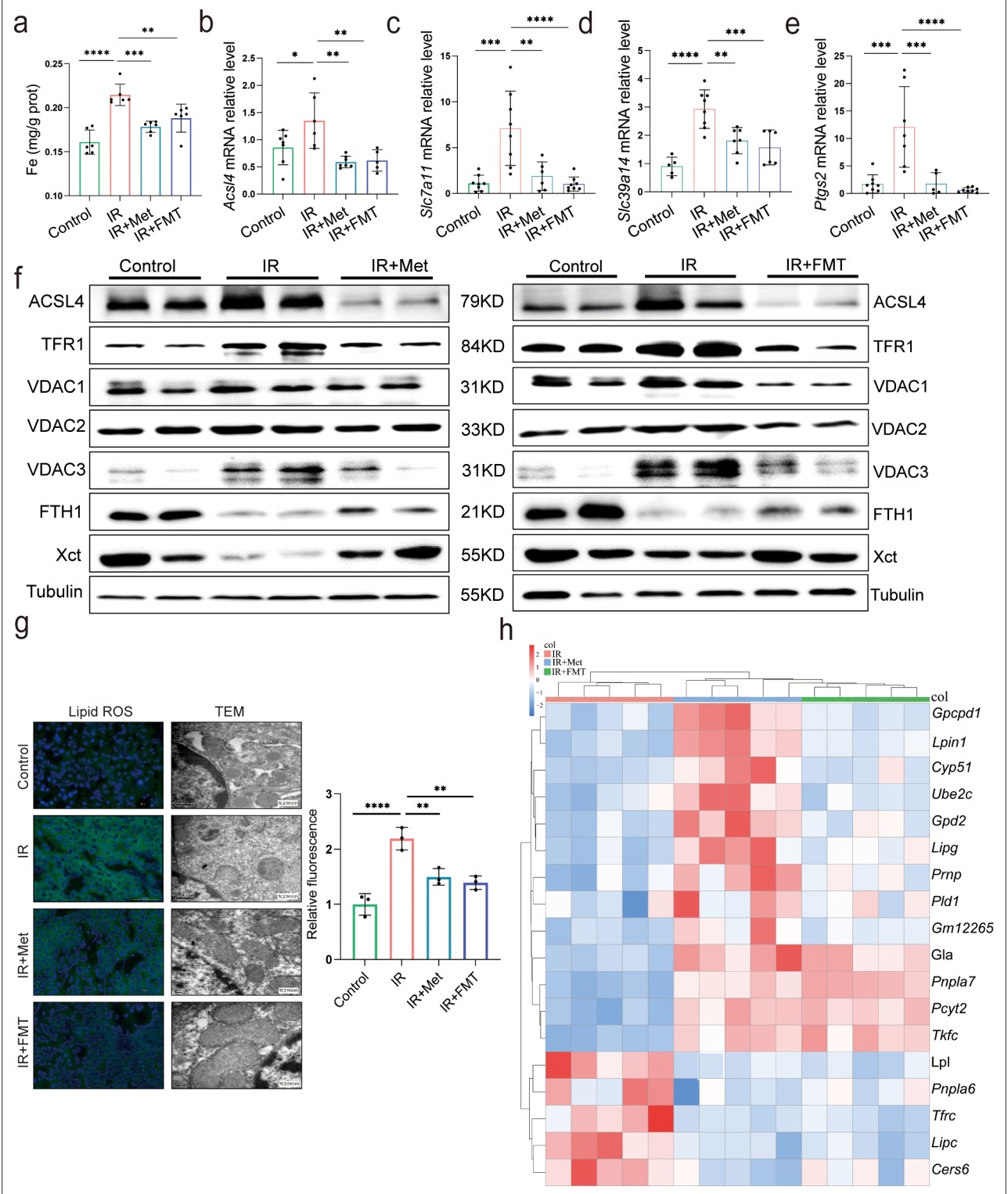

**Figure 3.** Metformin alleviates hepatic ischemia/reperfusion injury (HIRI) through inhibiting ferroptosis. (**a**) Levels of Fe in liver tissue ($n = 6$). Quantitative real-time PCR (qRT-PCR) analysis of *Acsl4* (**b**), *Slc7a11* (**c**), *Slc39a14* (**d**), and *Ptgs2* (**e**) ($n = 5$–8/group). (**f**) Western blot analysis of ACSL4, FTH1, VDAC1, VDAC2, VDAC3, TFR1, and Xct in liver tissue ($n = 3$/group). (**g**) Lipid reactive oxygen species (ROS) staining and transmission electron microscopy (TEM) analysis of liver tissues ($n = 3$/group). (**h**) Ferroptosis-related transcriptome cluster diagram heatmap. Data are expressed as mean ± standard deviation. One-way analysis of variance (ANOVA) was used to analyze statistical differences; *$p < 0.05$, **$p < 0.01$, and ***$p < 0.001$, $p$****$<0.0001$.

The online version of this article includes the following source data for figure 3:

*Figure 3 continued on next page*

*Figure 3 continued*

**Source data 1.** Original file for the Western blot analysis in *Figure 3f*.

**Source data 2.** PDF containing *Figure 3f* and original scans of the relevant Western blot analysis with highlighted bands and sample labels.

the apparent increase of anti-ferroptotic FTH1 and Xct levels (*Figure 3f*), The results were further consolidated by lipid reactive oxygen species (ROS) staining and transmission electron microscopy (TEM) analysis (*Figure 3g*). Additionally, transcriptome data on genes associated with anti-ferroptosis showed an increase in both IR + Met and IR + FMT groups, whereas proferroptic genes showed a decrease, when compared with IR group (*Figure 3h*). Collectively, the results suggested that FMT performed protection against HIRI by inhibiting ferroptosis.

## Metformin induces GABA-producing gut microbiota

To elucidate the molecular mechanisms through which pathway participates metformin-treated IR injury, we analyzed gene expression profiles of each group mice. Transcriptome sequencing analysis revealed that 9697 genes were in common among four groups (*Supplementary file 7*). Therefore, we used these common genes for KEGG(Kyoto Encyclopedia of Genes and Genomes) analysis, showing that similar mRNA changes are mainly concentrated in the three top pathways: lipid metabolism, carbohydrate metabolism, and amino acid metabolism (*Figure 4a*). Given the close relevance between lipid metabolism and ferroptosis (*Supplementary file 4*), and the fact of carbohydrate metabolism is a primary way to metabolize amino acids, 22 species of amino acid were detected in liver tissues using high-performance liquid chromatography with tandem mass spectrometric detection (HPLC–MS/MS) for further identification of key metabolites involved in the role of metformin against HIRI-induced ferroptosis. It was found that only GABA level is significantly increased by metformin treatment and FMT treatment (*Supplementary file 5*), further verifying by the data of ELISA detection (*Figure 4b*). Moreover, it was observed that the genus of *Bacteroides* had a significant increase based on the 16S rRNA gene sequencing of metformin-treated mice microbes. qRT-PCR analysis of eukaryotic GABA synthesis key enzymes including *Aldh5a1*, *Abat*, *Gad1*, and *Gad2* in liver tissues were not upregulated by metformin (*Figure 4d*), which was also confirmed by Western blotting detection of GAD1 and GAD2 in liver (*Figure 4f*). Interestingly, the fecal GABA concentration was shown a significant increase in metformin-treated mice, which was copied by the IR + FMT group, but eliminated by antibiotics treatment (*Figure 4e*). In order to further identify the increased GABA originates from gut microbiota, two key enzymes of prokaryotes GABA synthesis, glutamic acid decarboxylase (GAD) and putrescine aminotransferase (PAT), were detected on DNA level, finding that both of them are significantly increased in the feces from IR + Met and IR + FMT groups (*Figure 4g, h*). In order to clarify the specific effects of metformin on microbiota, given the big safety margin, healthy volunteers were recruited for a 1 week of daily oral 500 mg dose of metformin trial. Fecal samples were collected before and after oral administration of metformin for metagenomic analysis. Detailedly, the species of *Bacteroides* containing *Bacteroides thetaiotaomicron*, *Bacteroides unifomis*, and *Bacteroides salyersiae* were enriched in human gut after metformin administration (*Figure 4i*). Excitingly, the increase of GABA production, and the expression of GAD and PAT were shown in the results of fecal experiments in vitro and in vivo, accompanied by the decreased glutamic acid (Glu), a key substrate for gut microbiota to synthesize GABA (*Figure 4j, k*). Besides, putrescine, another substrate of GABA synthesis, was not found in primary culture medium but appeared after 24 hr fermentation and then was consumption later in the anaerobic culture of human feces experiment (*Figure 4l*). Taken collectively, GABA-producing gut microbiota was upregulated by metformin.

## GABA is the critical metabolite of metformin-reshaped gut microbiota against HIRI-induced ferroptosis

The animal experiment was designed to further observe the effects of GABA on HIRI-induced ferroptosis (*Figure 5a*). As expected, the data showed that GABA treatment reduced histological damage and liver glycogen consumption (*Figure 5b*). Serum liver function detection showed that ALT and AST activities were dramatically depressed after GABA treatment (*Figure 5c*). The decreased MDA and Fe as well as increased GSH were observed in further determination, confirming the protection of GABA on liver (*Figure 5d*). The results of Western blotting showed the anti-ferroptosis effect of GABA

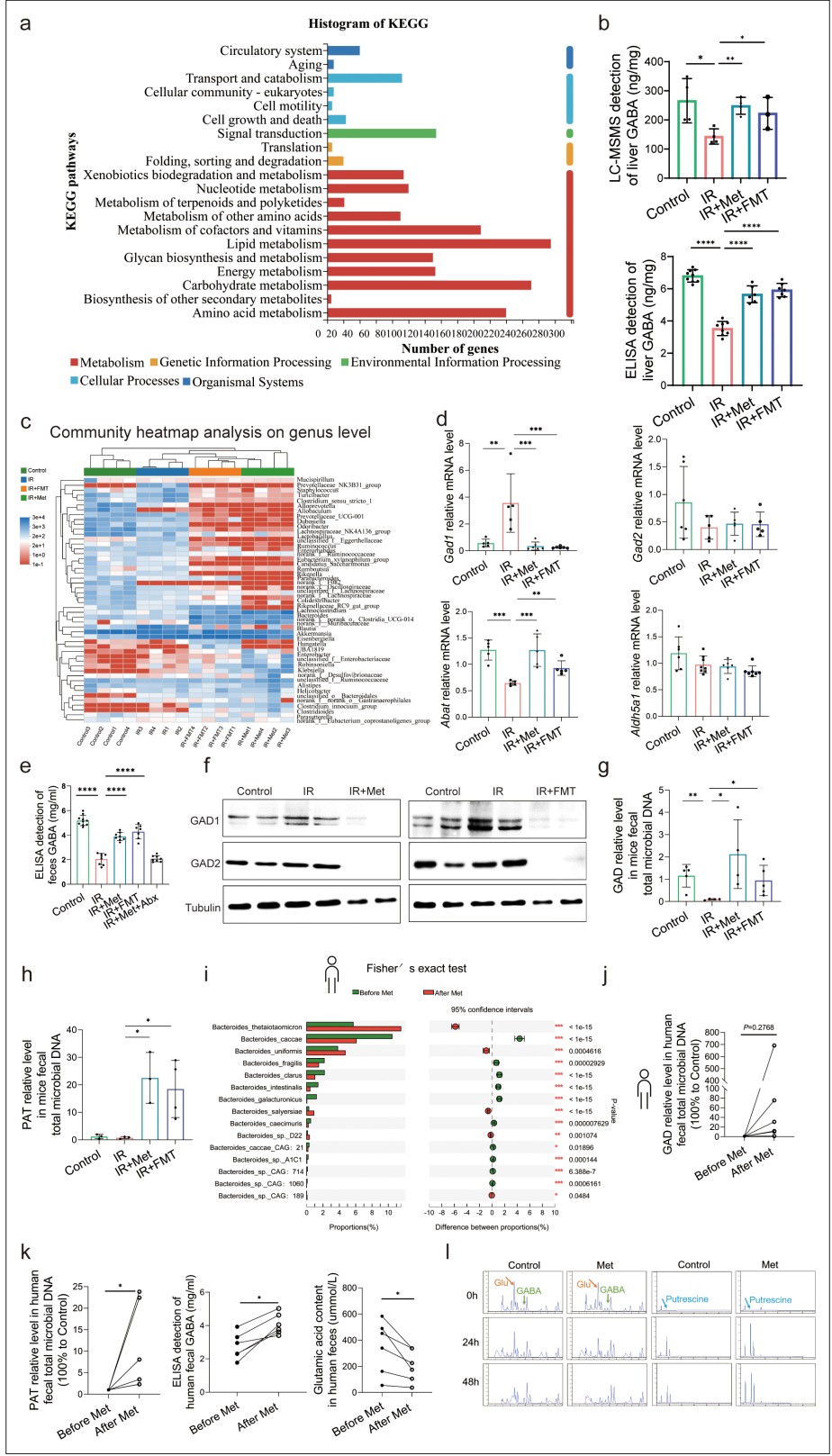

**Figure 4.** Metformin induces gamma-aminobutyric acid (GABA)-producing gut microbiota. (**a**) KEGG analysis of transcriptome. (**b**) High-performance liquid chromatography with tandem mass spectrometric detection (HPLC–MS/MS) of GABA content in liver, ELISA of GABA in liver (*n* = 6–9/group). (**c**) Community heatmap analysis on genus level. (**d**) Quantitative real-time PCR (qRT-PCR) analysis of key metabolite enzyme glutamate decarboxylase

*Figure 4 continued on next page*

*Figure 4 continued*

1 (*Gad1*), glutamic acid decarboxylase 2 (*Gad2*), 4-aminobutyrate aminotransferase (*Abat*), and aldhehyde dehydrogenase family 5 (*Aldh5a1*) in liver (*n* = 5–8/group). (**e**) ELISA of GABA in feces (*n* = 7–10/group). (**f**) Western blot analysis of GAD1 and GAD2 in liver. (**g, h**) qRT-PCR analysis of glutamic acid decarboxylase (GAD) and putrescine aminotransferase (PAT) in mice feces (*n* = 3–5/group). (**i**) Metagenomic sequencing analysis of human feces. (**j, k**) The level of GAD, PAT, GABA, and Glu in human feces (*n* = 5–6/group). (**l**) The level of GABA, Glu, and putrescine in human feces vitro culture (*n* = 3/group). Data are expressed as mean ± standard deviation. One-way analysis of variance (ANOVA) was used to analyze statistical differences; *p < 0.05, **p < 0.01, and ***p < 0.001, p****<0.0001.

The online version of this article includes the following source data for figure 4:

**Source data 1.** Original file for the Western blot analysis in *Figure 4f*.

**Source data 2.** PDF containing *Figure 4f* and original scans of the relevant Western blot analysis with highlighted bands and sample labels.

treatment by the significant reduction in ACSL4, TFR1, VDAC1,2,3 and increase of anti-ferroptotic FTH1 levels, which had the same attenuating effect as metformin and FMT treatment (*Figure 5e*). In addition, the clustergram showed GABA upregulated the genes of anti-ferroptosis and depressed the proferroptic ones, which was similar to the effect after metformin treatment (*Figure 5f*). In summary, GABA treatment reduced HIRI-induced ferroptosis.

## Discussion

Metformin has been well researched for treating various liver diseases due to its multifarious effects. Notably, it has been well documented that oral administration of metformin lead to therapeutic effects on HIRI (*Li et al., 2020*), leaving a thought-provoking question in its underlying mechanism. In the current study, the changes in the gut microbiota altered by metformin are demonstrated to be necessary and sufficient for conferring liver injury protection. Moreover, it has been confirmed that metformin attenuated HIRI by the increased GABA from reshaped gut microbiota, through anti-ferroptosis.

Increasing evidence shows that the gut microbiota is critical for the pleiotropic actions of metformin (*Milosevic et al., 2019*; *Forslund et al., 2015*; *Li et al., 2022*). Interestingly, metformin was previously found to have a much higher concentration in the gut lumen than other organs (*McCreight et al., 2016*), as also confirmed by the current study. In fact, that concentration favored the action of metformin on reshaping gut microbes. Despite the distinct heterogeneity of metformin-reshaped gut microbiota in recent research, *Bacteroides* with a high abundance of gut microbes was reported to be reduced after metformin treatment in studies of hyperlipidemia and diabetes (*Sun et al., 2018*; *Zhang et al., 2021*), which other studies vary from for the different action of this bacteria (*Lee et al., 2018*; *Shi et al., 2021*). In our study, we used a disease model of HIRI, which may have unique characteristics compared to other disease models. It is possible that the specific disease model influenced the response of the gut microbiota. Additionally, the starting microbiota of the recipients and the characteristics of the donor microbiota used for FMT could also play a role in the disparity. Our findings showed an increase abundance of the genus of *Bacteroides* after metformin treatment supported by the data of mice and healthy volunteers. Detailedly, the increased *Bacteroides* especially *B. thetaiotaomicron*, *B. unifomis*, and *B. salyersiae* were observed in the human gut after metformin administration. The enriched *B. thetaiotaomicron*, a glutamate-fermenting commensal, was reported to be decreased in obese individuals and was inversely correlated with serum glutamate concentration (*Liu et al., 2017a*). Therefore, *B. thetaiotaomicron* might be the potential bacterium regulating of GABA production. Little research has been done on *B. unifomis* and *B. salyersiae*, needing a further exploration of their characteristics in the future. Notably, *Phascolarctobacterium*, coexistly with *B. thetaiotaomicron* (*Ikeyama et al., 2020*) was reported as an emerging probiotic which showed a significant increase in our study. Taken collectively, those reshaped bacteria after metformin treatment may be responsible for attenuating HIRI.

Increased GABA was shown from the metformin-reshaped gut microbiota for the first time, verified by our results of feces from mice and humans. As the critical metabolite of bacteria against acid stress, GABA was extensively produced by gut microbes, especially the species with strong capacity

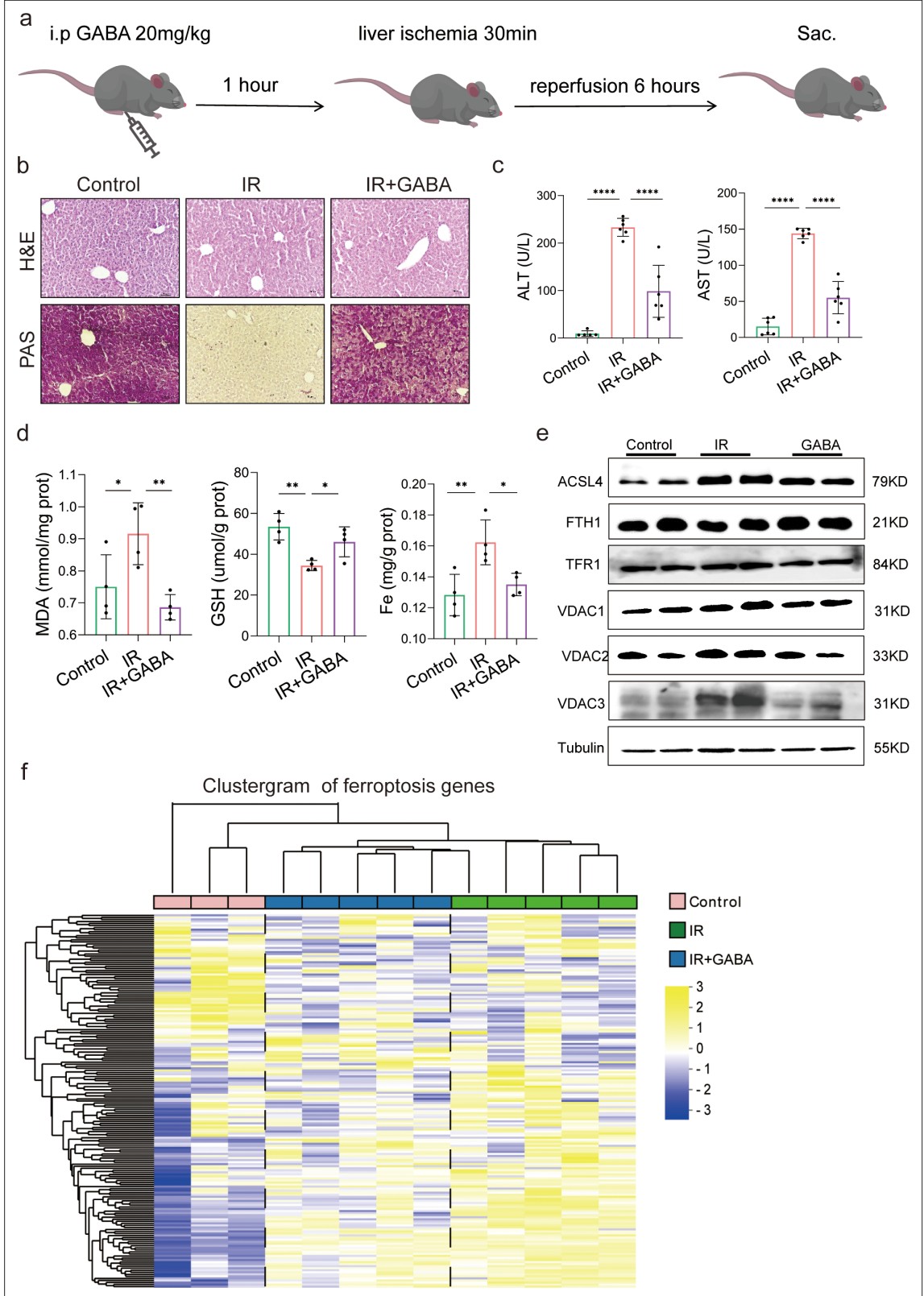

**Figure 5.** Gamma-aminobutyric acid (GABA) is the critical metabolite of metformin-reshaped gut microbiota against hepatic ischemia/reperfusion injury (HIRI)-induced ferroptosis. (**a**) Diagram of GABA treatment experiment on mice model. (**b**) Hematoxylin–eosin (H&E) and periodic acid-Schiff (PAS) staining of liver (*n* = 3/group). (**c**) Serum alanine transaminase (ALT) and aspertate aminotransferase (AST) levels (*n* = 6/group). Levels of malondialdehyde (MDA), glutathione (GSH), and Fe in liver tissues (**d**) (*n* = 4/group). (**e**) Western blot analysis of ASCL4, FTH1, TFR1, and VADC1,2,3 in

*Figure 5 continued on next page*

_Figure 5 continued_

liver (_n_ = 4/group). (**f**) Ferroptosis-related transcriptome cluster diagram heatmap (_n_ = 3–5/group). Data are expressed as mean ± standard deviation. One-way analysis of variance (ANOVA) was used to analyze statistical differences; *p < 0.05, **p < 0.01, and ***p < 0.001, p****<0.0001.

The online version of this article includes the following source data for figure 5:

**Source data 1.** Original file for the Western blot analysis in _Figure 5e_.

**Source data 2.** PDF containing _Figure 5e_ and original scans of the relevant Western blot analysis with highlighted bands and sample labels.

of acid resistance. Interestingly, our analysis of the sequences in PUBMED Gene bank found that those three species of _Bacteroides_, _B. thetaiotaomicron_, _B. unifomis_, and _B. salyersiae,_ which belong to metformin-reshaped microbiota, contain the key enzyme genes of GABA synthesis GAD and PAT (_Supplementary file 6_), further confirmed by the qRT-PCR result of human fecal DNA in vivo and in vitro experiment. Specifically, the species with the strong ability of producing GABA such as _B. thetaiotaomicron_, _B. unifomis_, and _B. salyersiae_ were increased in metformin-treated volunteers. Some studies were published to elucidate the mechanism of metformin regulating bile acids and folate metabolite by reshaped gut microbiota, however, there is no report about how metformin to affect GABA synthesis of gut microbes. The data from anaerobic cultured human feces showed that metformin promoted the increase of GAD and PAT, and GABA level along with the substrate consumption of glutamate and putrescine in a time-dependent manner.

GABA production was reported to be critical against acid stress for microbiota to survive and preserve metabolic activity, especially in the human gut with a lower pH than other animals (_Otaru et al., 2021_). Humans benefit from bacteria-derived GABA. Peripheral GABA was first demonstrated in intestinal flora in 1981 and further confirmed by recent studies (_Schafer et al., 1981_). In addition to the important role in the central neural system and immune system, GABA showed a protective effect in different acute liver injury models (_Wang et al., 2017_; _Shilpa et al., 2013_). Published papers have shown that GABA performs beneficial effect through binding to the GABA receptors or promoting histone acetylation (_Clark et al., 2021_; _Zeng et al., 2019_). Our data from GABA-treated mice showed protective effects against HIRI-induced ferroptosis. However, further studies are required to examine the exact molecular mechanisms of GABA regulating ferroptosis.

In conclusion, we have demonstrated metformin-reshaped gut microbiota performs therapeutic responses for HIRI. Gut microbiota has been obviously modified while receiving oral metformin treatment, suggesting that metformin-reshaped gut microbiota has enhanced intestinal mucosal function so that reduce harmful substances to the liver, which most efficiently evade damage. Specifically, GABA-producing bacteria were significantly regulated after metformin administration. Demonstrated for the first time, we showed that the increased GABA is a key metabolite of gut microbiota for metformin to mitigate HIRI. These results offer a unique therapeutic strategy, with optimized gut microbiota, that can serve as a target for metformin treating liver diseases.

# Materials and methods

## Animals

Six- to eight-week-old specific pathogen-free male C57BL/6 mice were purchased from Beijing Weitonglihua Experimental Animal Technology Co Ltd (Beijing, China). All mice were housed under controlled temperature and humidity conditions with a 12-hr light–dark cycle and free access to food and water. The mice were fasted overnight before the experiments. All animal experiments were approved by the Institutional Animal Care and Use Committee of the Experimental Animal Centre of Wenzhou Medical University (wydw2021-0348).

## Healthy volunteers

All research was conducted in accordance with both the Declarations of Helsinki and Istanbul, all research was approved by the Ethics Committee of the Second Affiliated Hospital of Wenzhou Medical University (2022-K-08-01), and written consent was given in writing by all subjects.

## Hepatic IR mouse model

A mouse model for HIRI was established as previously described (*Liu et al., 2019*). Briefly, mice were anesthetised with pentobarbital sodium. A noninvasive microvascular artery clip was placed on the left branches of the hepatic artery and portal vein for 30 min, and the clip was removed for reperfusion for 6 hr. During the study period, body temperature was maintained at 37°C using a heating lamp.

## Experimental design

Mice were randomly divided into the several groups. In general, the arterial and portal vessels to the cephalad lobes were clamped for 30 min and reperfusion for 6 hr. No vascular occlusion in sham-controlled mice. In the treatment groups, animals were treated with metformin (1 mg/ml), metformin modified FMT or metformin (1 mg/ml) mixed with antibiotics (1 mg/ml penicillin, 1 mg/ml neomycin sulfate, 1 mg/ml metronidazole, and 0.16 mg/ml gentamicin), in the drinking water for 1 week. Specifically, FMT-treated mice were fed with washed metformin-treated mice feces, which had been detected by HPLC showing that there was no metformin left in the feces. Other two groups of mice were infused at 1 hr prior to the onset of liver ischemia with a single dose of deferoxamine (DFO) (20 mg/kg i.p.) or GABA (100 mg/kg i.p.), dissolved in phosphate-buffered saline (PBS). Mice were sacrificed after reperfusion; liver and serum samples were collected for analysis.

## Measurement of serum ALT, AST level, liver Fe content, and GABA level

Serum AST and ALT levels were measured using assay kits (Nanjing Jiancheng Bioengineering Institute, C010-2-1, C009-2-1). Liver samples were homogenized in saline, and Fe concentrations were measured by tissue Fe assay kit (Nanjing Jiancheng Bioengineering Institute, A039-2-1). GABA level was detected by an ELISA kit (Wuhan MSKBIO, KT21124).

## Measurement of lipid peroxidation, GSH, and superoxide anion levels of liver tissues

Malondialdehyde (MDA) Detection Kit (Nanjing Jiancheng Bioengineering Institute, A003-1-2) was determined to select the MDA level as a marker of lipid peroxidation. The assay was performed according to the manufacturer's instructions. Lipid ROS was measured by BODIPY 581/591 C11 (MKbio, 217075-36-0). Sections were stained with BODIPY 581/591 C11 and nucleated with 4',6-diamidino-2-phenylindole (DAPI) (Beyotime Institute of Biotechnology, A0131). Images were acquired under a fluorescence microscope (Nikon, Tokyo, Japan). The liver was homogenized, and the supernatant was collected for GSH analysis using a GSH assay kit (Nanjing Jiancheng Bioengineering Institute, A006). Frozen sections of the liver (8 µm) were placed on glass slides and incubated with 10 mmol/l DHE (Beyotime Institute of Biotechnology, S0063) in a dark container at 37°C for 30 min. Sections were rinsed three times in PBS then observed using an upright microscope (Nikon, Tokyo, Japan).

## H&E and PAS staining

For histopathological examination, liver and intestinal tissues were fixed in 4% paraformaldehyde overnight. Selected tissue blocks were processed using a routine overnight cycle in a tissue processor. Tissue blocks were then embedded in wax and serially sliced into 5 µm sections. H&E and PAS staining performed as instructions provided by the reagent manufacturer (Solarbio, G1120, G1281). Visualization and images were obtained under an optical microscope (Nikon, Tokyo, Japan).

## Histoimmunofluorescence staining

During immunofluorescence, sections were dyeing with primary antibody (1:100) Occludin (Proteintech, 27260-1-AP) and zonula occludens 1 (ZO-1) (Proteintech, 66378-1-IG) and incubated at 4°C overnight. Secondary antibody coupled with Alexa Fluor 488 (Bioss, bs-40296G-AF488) was incubated at 37°C for 1 hr. After additional PBS washing, the sections were sealed with an antifluorescence quencher containing DAPI. Immunofluorescence images were obtained using a laser scanning confocal microscope (Nikon, Tokyo, Japan) and quantified by ImageJ (NIH, USA).

## TEM analysis

After reperfusion, the mice were euthanized, and the livers were excised and washed with precooled PBS (pH 7.4). Part of the liver was then removed and incubated overnight in 0.1 M PBS (pH 7.4) containing 2.5% glutaraldehyde. The target tissues were cut into 50-µm-thick sections using a vibratome. Selected areas of the livers were postfixed in 1% osmium tetroxide for 1 hr, dehydrated in a graded ethanol series and embedded in epoxy resin. The polymerization was performed at 80°C for 24 hr. Ultrathin sections (100 nm) were cut, stained with uranyl acetate and lead citrate and viewed under a JEM2000EX TEM (JEOL, Tokyo, Japan). Five fields were randomly selected for each sample to examine mitochondria with ferroptosis features.

## qRT-PCR analysis

Total RNA was extracted from liver and intestinal tissues using TRIzol (Yamei, YY101). Isolated RNA was reverse transcribed to cDNA using a kit (Vazyme, R323-01). The obtained cDNA was subjected to PCR using primers designed to detect Interleukin (*Il)6*, *Il1b*, *Il18*, recombinant solute carrier family 7, Member 11 (*Slc7a11*), solute carrier family 39, member 14 (*Slc39a14*), acyl-CoA synthetase long-chain family member 4 (*Acsl4*), cyclooxygenase-2 (*Ptgs2*), *Tjp1*, *Ocln*, *Cldn7*, Junctional adhesion molecule 1 (*F11r*), Junctional adhesion molecule 4 (*Igsf5*), succinic semialdehyde dehydrogenase (*Aldh5a1*), GABA transaminase (*Abat*), *Gad1*, *Gad2*, and *Actb* (primer sequences are listed in *Supplementary file 1*). Gene expression was determined using the SYBR Green kit (Vazyme, Q711-02). All the results were normalised against β-actin expression using the Thermal Cycler Dice Real Time System (ABI QuantStudio6, Singapore).

## Western blotting

Total protein samples were extracted from tissues using RIPA(radioimmunoprecipitation assay) lysis buffer (Yamei, PC101). Protein concentrations were determined using a BCA protein detection kit (Yamei, ZJ101). Proteins were separated using 10% sodium dodecyl sulfate–polyacrylamide gel electrophoresis and transferred to polyvinylidene fluoride membranes. Membranes were blocked in 5% skimmed milk then incubated with the primary antibodies ACSL4 (AbCam, ab155282), Transferrin Receptor 1 (TFR1) (Abclonl, A5865), ferritin heavy chain (FTH1) (AbCam, ab65080), Occludin (Proteintech, 27260-1-AP), GAPDH (Proteintech, 0494-1-AP), and β-tubulin (Proteintech, 10068-1-AP) at 4°C overnight. After washing three times with Tris-buffered saline, the membranes were incubated with appropriate anti-rabbit or anti-mouse secondary antibodies at room temperature for 1 hr. Imprinting was observed using chemiluminescence (Yamei, SQ201) and an Odyssey imaging system (Li-Cor Biosciences, NE, USA).

## HPLC–MS/MS detection of metformin and liver amino acids

Sample separation was performed using a ZORBAX Eclipse XDB-C18 column (4.6 × 150 mm, Agilent, USA) using an injection volume of 5 µl. Metformin detection conditions are as follows: column temperature of 30°C, one mobile phase is $NaH_2PO_4$ (Sigma, V900060) containing 5 mmol/l sodium dodecyl sulfonate (Sigma, V900859) with pH 3.5 while the other mobile phase is acetonitrile (Aladdin, A104440). The metformin content in each sample was measured by HPLC (Agilent 1260, USA). Amino acid detection conditions are as follows: column temperature of 40°C, mobile phase A of 10% methanol/water (containing 0.1% formic acid), mobile phase B of 50% methanol/water (containing 0.1% formic acid), and flow rate of 0.4 ml/min. Mass spectrometry was performed using a triple quadrupole mass spectrometer with an ESI(electrospray ionization) source in negative ionization mode. High-purity nitrogen was used as the nebulizing and drying gas. Quantification was performed in multiple reaction monitoring mode.

## 16S rRNA sequencing

16S rRNA amplicon sequencing was performed at Chunlab Inc (Seoul, Korea) with MiSeq system (Illumina). Briefly, for preparation of MiSeq library amplicons, target gene (16S rRNA V3–V4 region) was amplified using 338F (5'-ACTCCTACGGGAGGCAGCAG-3') and 806R (5'-GGACTACHVGG-GTWTCTAAT-3') primers, using an ABI GeneAmp 9700 PCR thermocycler (ABI, CA, USA). The PCR product was extracted from 2% agarose gel and purified using the AxyPrep DNA Gel Extraction Kit (Axygen Biosciences, Union City, CA, USA) according to the manufacturer's instructions and quantified

using a Quantus Fluorometer (Promega, USA). Normalization was performed the counts of individual OTUs(operational taxonomic units) in a sample by dividing the total counts of all OTUs within that sample followed by a multiplication by resulting in relative abundance expressed.

## Metagenomic sequencing

Total genomic DNA was extracted from human fecal samples as mentioned in 16S rRNA sequencing. DNA extract was fragmented to an average size of about 400 bp using Covaris M220 (Gene Company Limited, China) for paired-end library construction. Paired-end library was constructed using NEXT-flexTM Rapid DNA-Seq (Bioo Scientific, Austin, TX, USA). Adapters containing the full complement of sequencing primer hybridization sites were ligated to the blunt-end of fragments. Paired-end sequencing was performed on Illumina NovaSeq/Hiseq Xten (Illumina Inc, San Diego, CA, USA) at Majorbio Bio-Pharm Technology Co, Ltd (Shanghai, China) using NovaSeq Reagent Kits/HiSeq X Reagent Kits according to the manufacturer's instructions (https://www.illumina.com).

## Statistical analysis

GraphPad Prism version 9.0 (GraphPad Software, San Diego, CA) was used for statistical treatment. Experimental data were shown as the mean ± standard deviation. Two-tailed unpaired Student's *t*-test and one-way analysis of variance with Tukey's correction were used for all comparisons of mice-related experiments. A p-value <0.05 was considered significant. The sample distribution was determined using a Kolmogorov–Smirnov normality test.

## Acknowledgements

We appreciate the support of Suzhou Inhal Pharma Co, Ltd and Zhejiang Xiaolun Intelligent Manufacturing Co, Ltd. We also thank Shanghai Majorbio Bio-pharm Technology Co, Ltd for assisting with the 16S rRNA and metagenomics analysis, Suzhou Bionovogenes for helping with the HPLC–MS/MS analysis. This work was supported by the University Students Science and Technology Innovation Program, Zhejiang Province under Grant (2022R413C074); the Basic Public Welfare Research Project of Zhejiang Natural Science Foundation of China under Grant (LGF22H030011); Suzhou Inhal Pharma Co, Ltd under Grant (KJHX2212); and Zhejiang Xiaolun Intelligent Manufacturing Co, Ltd under Grant (KJHX2202).

## Additional information

### Competing interests

Xiujie Liu, Yuqing Ye, Jesse Zhu: Employee of Suzhou Inhal Pharma Co, Ltd. The other authors declare that no competing interests exist.

### Funding

| Funder | Grant reference number | Author |
| --- | --- | --- |
| Zhejiang University Student Science and Technology Innovation Activity Plan | 2022R413C074 | Wantie Wang |
| Zhejiang Province Public Welfare Technology Application Research Project | LGF22H030011 | Fangyan Wang |
| Suzhou Inhale Pharma Co, Ltd | KJHX2212 | Fangyan Wang |
| Zhejiang Xiaolun Intelligent Manufacturing Co, Ltd | KJHX2202 | Fangyan Wang |

The funders had no role in study design, data collection, and interpretation, or the decision to submit the work for publication.

## Author contributions

Fangyan Wang, Supervision, Validation, Writing – original draft; Xiujie Liu, Writing – original draft, Project administration; Furong Huang, Gaoyi Ruan, Data curation, Formal analysis; Yan Zhou, Conceptualization, Investigation; Xinyu Wang, Resources; Zhengyang Song, Zenglin Tan, Software; Sisi Wang, Dibang Shi, Methodology; Xiaoting Wang, Formal analysis; Xiawei Ji, Software, Formal analysis; Eryao Zhang, Data curation; Yuqing Ye, Jesse Zhu, Writing – original draft, Writing – review and editing; Chuang Wang, Project administration; Wantie Wang, Supervision, Funding acquisition

## Author ORCIDs

Fangyan Wang ⬥ http://orcid.org/0000-0003-0555-8495
Xiujie Liu ⬥ http://orcid.org/0000-0003-2230-3865
Wantie Wang ⬥ https://orcid.org/0000-0003-4129-5549

## Ethics

All research was approved by the Ethics Committee of the Second Affiliated Hospital of Wenzhou Medical University (2022-K-08-01). Written consent was given in writing by all subjects.
All animal experiments were approved by the Institutional Animal Care and Use Committee of the Experimental Animal Centre of Wenzhou Medical University (wydw2021-0348).

Reviewer #1 (Public Review): https://doi.org/10.7554/eLife.89045.4.sa1
Reviewer #2 (Public Review): https://doi.org/10.7554/eLife.89045.4.sa2
Author Response https://doi.org/10.7554/eLife.89045.4.sa3

# Additional files

## Supplementary files

- Supplementary file 1. Sequence list of relevant mouse genes.
- Supplementary file 2. Source list of antibiotics used in the article.
- Supplementary file 3. Data of ferroptosis inhibitor DFO treatment experiments.
- Supplementary file 4. Heatmap of ferroptosis gene cluster in each group.
- Supplementary file 5. Heatmap of amino acid content detection in each group.
- Supplementary file 6. Sequence list of *Bacteroides* common genes PAT and GAD.
- Supplementary file 7. Venn diagram of genes in each group.
- MDAR checklist

## Data availability

The original contributions presented in the study are publicly available. Sequencing data have been uploaded to NCBI including 16Ss sequencing (PRJNA1045386), transcriptome sequencing (PRJNA1048138), and metagenomic sequencing (PRJNA1049568). Other data can be found on Dryad: https://doi.org/10.5061/dryad.kprr4xhc4.

The following datasets were generated:

| Author(s) | Year | Dataset title | Dataset URL | Database and Identifier |
|---|---|---|---|---|
| Liu X | 2023 | 16s sequencing | https://www.ncbi.nlm.nih.gov/bioproject/PRJNA1045386/ | NCBI BioProject, PRJNA1045386 |
| Liu X | 2023 | transcriptome sequencing | https://www.ncbi.nlm.nih.gov/bioproject/PRJNA1048138/ | NCBI BioProject, PRJNA1048138 |
| Liu X | 2023 | metagenomic sequencing | https://www.ncbi.nlm.nih.gov/bioproject/PRJNA1049568/ | NCBI BioProject, PRJNA1049568 |

*Continued on next page*

*Continued*

| Author(s) | Year | Dataset title | Dataset URL | Database and Identifier |
|-----------|------|---------------|-------------|-------------------------|
| Liu X | 2024 | Gut microbiota-derived gamma aminobutyric acid from metformin treatment reduces hepatic ischemia/reperfusion injury through inhibiting ferroptosis | https://doi.org/10.5061/dryad.kprr4xhc4 | Dryad Digital Repository, 10.5061/dryad.kprr4xhc4 |

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
