## [Editor Report · eLife assessment]

This study presents a **valuable** finding on the impact of metformin-induced shifts in gut microbial community structure and metabolite levels for drug efficacy in a mouse model of liver injury. The current evidence supporting the claims of the authors is **solid**. This paper will be of broad interest to researchers across multiple disciplines, including the microbiome, liver disease, and pharmacology.

---

## [Referee Report · Reviewer #1 (Public Review)]

Many drugs have off-target effects on the gut microbiota but the downstream consequences for drug efficacy and side effect profiles remain unclear. Herein, Wang et al. use a mouse model of liver injury coupled to antibiotic and microbiota transplantation experiments. Their results suggest that metformin-induced shifts in gut microbial community structure and metabolite levels may contribute to drug efficacy. This study provides valuable mechanistic insights that could be dissected further in future studies, including efforts to identify which specific bacterial species, genes, and metabolites play a causal role in drug response. Importantly, although some pilot data from human subjects is shown, the clinical relevance of these findings for liver disease remain to be determined.

Comments on revised version:

The authors have now addressed my original concerns.

---

## [Referee Report · Reviewer #2 (Public Review)]

The authors examine the use of metformin in the treatment of hepatic ischemia/reperfusion injury (HIRI) and suggest the mechanism of action is mediated in part by the gut microbiota and changes in hepatic ferroptosis. The concept is intriguing and their results have potential to better understand the pleiotropic functions of metformin. The histological and imaging studies were considered a strength and reveal a significant impact of metformin post-HIRI. The connections with GABA producing bacteria adds to our understanding of the chemical signals exchanged between the host and microbiota. While the authors have characterized these connections in mice, how/if these observations translate to humans remains to be determined.

---

## [Author Response]

The following is the authors’ response to the previous reviews.

**Reviewer #1 (Recommendations For The Authors):**
Many of my specific issues have been addressed in the revision. However, the data shown in Reviewer Fig. 1 and 2 is not sufficiently described to assess it's reliability and these new data do not appear to have been integrated into the paper. A response that more clearly states how the manuscript has been revised to address the comments is necessary.

We appreciate the opportunity to respond to your updated comments on our manuscript. We carefully considered the feedback and made changes to address the specific issues raised.

In response to your question of insufficient description of the data shown in Reviewer Fig. 1 and 2, we would like to confirm that we have taken this feedback seriously. Supplementary data, including the information provided in Reviewer Figures 1 and 2, have been fully described and integrated into the body of the manuscript according to your request. We ensured that the reliability and significance of new data were clearly presented to enhance the overall synthesis of the manuscript.

We are grateful to your valuable feedback, which undoubtedly contributed to the refinement of our manuscript. We hope that the revised version meets the standards of the journal and look forward to the opportunity for further deliberation.

**Reviewer #2 (Recommendations For The Authors):**
Additional feedback from the reviewer:"I think the authors have been responsive to my previous comments. However, I cannot find this new data in the main text but rather only in the response to reviewers. New data should be incorporated into the main text not the supplement as the controls are important to consider alongside the treatment groups. Lastly, while the authors include BODIPY in their approaches, their results are not quantitative. My suggestion was to include this data in a quantitative manner not just the images. Lastly, I am still somewhat puzzled about the connection with GABA. The rationale for its selection other than it was significantly changed is not strong."

Thank you for providing us with the latest feedback. We appreciate the opportunity to address the specific concerns raised and provide a detailed response to each point.

(1) Incorporation of New Data into the Main Text:

We acknowledge the reviewer's comment regarding the incorporation of new data into the main text rather than solely in the response to reviewers. In response to this feedback, we have diligently revised the manuscript to ensure that the new data, including controls, is now seamlessly integrated into the main body of the text. This modification allows for a more comprehensive and contextual presentation of the data, as recommended by the reviewer.

(2) Quantitative Presentation of BODIPY Results:

We understand the importance of presenting quantitative data for the BODIPY results, and we appreciate the reviewer's suggestion to include this information in a quantitative manner, not just as images. In line with this valuable feedback, we have revised the relevant sections to incorporate quantitative data alongside the images, providing a more robust and comprehensive presentation of the results.

(3) Rationale for the Selection of GABA:

In the present study, in order to elucidate the molecular mechanisms through which pathway participates metformin-treated IR injury, we analysed gene expression profiles of each group mice, showing that similar mRNA changes are mainly concentrated in the three top pathways: lipid metabolism, carbohydrate metabolism, and amino acid metabolism. Given the close relevance between lipid metabolism and ferroptosis, and the fact of carbohydrate metabolism is a primary way to metabolize amino acids, 22 species of amino acid were detected in liver tissues using HPLC-MS/MS for further identification of key metabolites involved in the role of metformin against HIRI-induced ferroptosis. It was found that only GABA level is significantly increased by metformin treatment and FMT treatment, further verifying by the data of ELISA detection. Consequently, we identified GABA was the main metabolism of metformin protecting from HIRI and focus on the source of GABA generation.

We would like to express our gratitude to your thorough evaluation and constructive feedback, which has undoubtedly contributed to the improvement of our manuscript.